# Perceived Health and Earnings: Evidence from the European Working Conditions Survey 2015

**DOI:** 10.3390/ijerph19010594

**Published:** 2022-01-05

**Authors:** Amaya Erro-Garcés, Maria Elena Aramendia-Muneta, María Errea, Juan M. Cabases-Hita

**Affiliations:** 1Institute for Advanced Research in Business and Economics (INARBE), Public University of Navarre (UPNA), 31006 Pamplona, Spain; elena.aramendia@unavarra.es (M.E.A.-M.); jmcabases@unavarra.es (J.M.C.-H.); 2Independent Researcher, 31008 Pamplona, Spain; maria.errea85@gmail.com

**Keywords:** perceived health, earnings, European comparison, income inequality, health policy

## Abstract

This paper aims to analyse the relationship between perceived health and earnings across Europe. Empirical analysis is based on the last published round from the European Working Conditions Survey (N = 43,850) and offers updated evidence on the effect of earnings on perceived health in 35 countries. The main findings show a positive and significant relationship between earnings and health, which is consistent with the existing literature. Moreover, health seems to be U-shaped relative to earnings. On the other hand, age is negatively related to health, which is consistent with previous research. This paper shows the health differences between countries, where cultural, geographic, and economic differences imply health inequalities across countries. From a practical perspective, understanding the dynamics of perceived health and earnings’ processes can contribute to health policy.

## 1. Introduction

Health status is closely related to employment conditions. Job characteristics or the nature of work may put health at risk (e.g., working too many hours or being exposed to chemical products in the workplace), but they may also have a positive impact on health [1]. Studies have often identified the powerful effects of income on health (the impact measured through changes in the mentioned variables), even after adjusting for socio-economic status, or when comparing self-reported health with some baseline health status [2,3,4].

Health status depends, not only on healthy habits, but also on income. The demand for health and healthcare, published by Grossman [5,6], showed the effects of education, wages, and age on health. Evidence suggests that working could be helpful for health improvement, when the value of the negative effects of unemployment exceeds the value of the negative effects of being employed [7,8]. Working can improve health when higher earnings result in healthier habits, in access to better healthcare services and/or in other socio-economic situations that positively affect health. Higher earnings are frequently associated with more or longer days at work [9]. However, evidence also shows an association between longer working days (the number of working hours is used in the literature as a working effort measure) and stress, a condition that has been demonstrated to have a negative impact on health [10,11].

The current paper aims to go one step further. It focuses on the effect of earnings on health for a population of active workers, considering microeconomic and macroeconomic factors by exploring the interaction between individual factors, occupation characteristics and work environment factors in 35 European countries.

## 2. Methods

### 2.1. Data

Data from the sixth round of the European Working Conditions Survey (EWCS), gathered from September to December 2015, have been used. This questionnaire collected demographic data and a broad range of factors regarding current working conditions. The specificity of this dataset is that it includes a population of active workers only. Retired or unemployed individuals were not considered. Working conditions evaluated in the survey include physical and psychosocial risk factors, employee participation, learning factors, gender issues, work environment, and health risks perceived at work. This EWCS 2015 data is complemented with Eurostat macroeconomic indicators [12]. These are used to incorporate countries’ contextual factors such as Gross Domestic Product (GDP) per capita and unemployment rates.

Among the 35 countries included, 28 are EU state members, and the remaining are Norway, Switzerland, Albania, the former Yugoslav Republic of Macedonia, Montenegro, Serbia and Turkey. In total, our sample comprises N = 43,850 individuals. Each country interviewed a different number of individuals, which ranged from 1000 to 3364 (see Figure 1 below).

### 2.2. Variables Description

#### 2.2.1. Health Assessment (H)

The level of Health assessed (H), refers to the respondent’s own assessment of his or her health. The European Statistics of Income and Living Conditions (EU-SILC) asks a similar question. The question in the EWCS 2015 used in this study is ‘How is your health in general? Would you say it is …?’. Five categories of response were included, from 1—Very good to 5—Very bad. This variable is inverted for our analysis so that the greater its value the better the individual’s health assessment.

The independent variables in Table 1 are divided into the five previously-mentioned categories: individual factors (I), job characteristics (JC), work environment (WE), macroeconomic factors (M) and earnings (E).

#### 2.2.2. Individual Factors (I)

The set of individual factors is composed of the following socio-demographic variables: gender, level of education, age, number of hours educating children/grandchildren, type of occupation, reporting having suffered discrimination due to ethnicity, marital status. Existing literature uses these variables for explaining health [13].

Nine different levels of education are considered. To homogenise data from different European countries, the International Standard Classification of Education (ISCED) is used to classify workers. Employees are divided into nine groups according to the highest level of education or training that they report to have successfully completed.

Additionally, the International Standard Classification of Occupations (ISCO) is applied to categorise occupations. The ISCO is divided into ten groups, but our sample presents only nine groups (managers, professionals, technicians and associate professionals, clerical support workers, service and sales workers, skilled agricultural, forestry and fishery workers, craft and related trades workers, plant and machine operators, assemblers and elementary occupations), because armed forces are omitted.

#### 2.2.3. Job Characteristics (JC)

Variables such as the size of the business or the increase in the number of hours an employee has to work since the job started are included in this category. The size of the company is used as a control variable, with four categories: self-employment/one employee, small firms (from 2 to 9 workers), companies characterized by medium size (10 to 249 employees) and the largest enterprises (250 employees or over). The group of smallest firms (those with one employee) is considered as the baseline. The increase in the number of hours at work is measured on a Likert scale from one to five, with one reflecting that hours had increased significantly since the individual started the job, and five reflecting that these had significantly decreased.

#### 2.2.4. Work Environment (WE)

The groups of variables that were included to represent the work environment are non-tangible statements related to climate at the workplace. Each statement is measured in a Likert scale (1—strongly disagree to 5—strongly agree). These are questions such as if employees feel appreciated when they do a good job, the degree to which employers trust employees, if conflicts are solved in a fair way, if the work is fairly distributed and if relations with other workers are good.

Questions related to health and safety at the workplace are also included in this category. This is used as a risk at work measure because it provides information regarding the positive or negative impact of the workplace on health (baseline level used is ‘no impact’. Dichotomous variables (1 = Yes; 0 = No) are used to measure these situations.

#### 2.2.5. Macroeconomic Factors (M)

The GDP per capita and unemployment rate, at a national level and from Eurostat statistics, are included as macroeconomic indicators. To facilitate comparisons between countries, GDP is measured at purchasing power parity (PPP). A dichotomous variable to distinguish when the GDP/unemployment of a country are above the median versus when its GDP/unemployment is below the median has also been created. Because the country GDP per capita, the unemployment rate of a country and the country are linearly dependent, including more than one of these variables would result in a mis-specified model due to perfect multicollinearity. Therefore, separate models are estimated.

#### 2.2.6. Earnings (E)

Monthly earnings are introduced in the model as a dependent variable. Monthly earnings squared are also included to test for a non-linear association between earnings and health.

## 3. Model Specification

The following equation is specified:*Pr(H_i_ = h | x_k_) = f(x_ik_β) + ε_i_*(1)
where *i* represents each worker, *h* is the possible level of self-assessed health (very bad, bad, fair, good and very good) and *k* is the number of independent variables or inputs included in the model. The explanatory variables (*x*) are individual factors (I), job characteristics (JC), work environment (WE), macroeconomic factors (M) and earnings (E), similarly to Wang et al. [14]. Robust standard errors are assumed in all the estimations.

Equations were estimated stepwise to facilitate the analysis of the association between earnings and health, controlling for other variables that may affect this association. A ‘conventional’ health equation was estimated first, which included individual socio-demographic information and job characteristics, work environment and macroeconomic factors only. Then, the macroeconomic variables (GDP and unemployment rate), country dummies and interactions were included in the regression model, separately. Table 2, Table 3 and Table 4 show the regression model results for the full models given that the stepwise estimation did not show inconsistencies. Standardized (beta) coefficients are reported to have a homogeneous measure of how much each variable associates with the health assessed, and to facilitate comparisons.

## 4. Results

### 4.1. Descriptive Statistics

Summary statistics are presented in Table 1. With an average health of 4.01, workers in our sample report good health. The sample is equally distributed by gender, with a 50.4% of respondents being men. The mean age of workers is 43.37 years old (SD = 12.75). Forty-one percent of the sample completed upper secondary education level. Among the rest of the workers surveyed, 13.4% finished lower secondary education studies, 13.1% complete bachelor education, 7% post-secondary education and 9.4% completed short cycle tertiary education level. In the extremes of the educational classification, in the lower levels, less than 5% of the workers only finished primary education and less than 1% early school, whereas in the upper levels, 9.3% of the employees obtained a master’s level education, and 1% a doctorate level education.

The most common occupations among the surveyed workers are service and sales workers (21.7%) and professionals (17.7%), followed by craft (11.8%) and technicians (11.2%), elementary occupations (10.4%), clerical support workers (8.6%), plant and machine operators (6.8%), managers (6.3%) and skilled agricultural (4.8%). Most employees that participated in the survey worked in medium and large firms (36.1% in businesses with 10 to 249 workers and 26.8% in the group of largest companies). Because the average level of hours increased at work since the job started is 2.85 (SD = 0.68), it can be concluded that most workers did not dramatically change the number of hours they work. Regarding the work environment, all variables present average scores higher than 3.75, which means that these ascribed behaviours occur habitually in the companies of the surveyed workers. The variable that measures if the individual thinks there is cooperation between colleagues at work is the best valued.

The mean GDP per capita (PPP) observed is EUR 30,000, and the mean rate of unemployment is 11.9%. The average monthly earnings considered in the sample is EUR 1346.01 (SD = 2278.87).

### 4.2. Regression Results

Regression results from estimating probit models are presented in Table 2, Table 3 and Table 4.

As shown in the three models, men have a greater probability of having very good health, while women are more likely to have a fair level of health assessed compared to the opposed gender. Age appears to be negatively associated with health, increasing the likelihood of bad or very bad health, and decreasing the likelihood of very good health. The number of hours caring for children is also associated with a greater probability of good health. The greater the level of education, the greater the association (and more significant) with the probability of health assessed being very good compared to the lowest educated group, early childhood education. The probability of bad health is also significantly associated with education, with the lowest educated group being more likely than any other group to report bad health. Regarding the type of occupation, compared to elementary workers, managers have a lesser probability of very good health being assessed. Technicians, clerical, or craft workers, though, are significantly more likely than elementary workers to report very bad health. Technician and clerical workers are, however, significantly less likely than the baseline group to report a good level of health.

Compared to self-employed individuals, workers at small firms are more likely to report higher health. The probability of a good self-assessed health is also greater for workers in bigger companies of 250 employees or over, but the estimate indicates the difference with self-employed is not as high as it is for employees of small companies. Regarding the interaction between occupation and age, results show how older workers, except for skilled agricultural and craft workers, have a greater probability, compared to elementary workers, of reporting a good or very good level of health.

Interestingly, the change in the number of working hours since the job started is not negatively associated with good or bad health reported, per-se. However, there are other work environment factors that are significantly associated with the health assessed. For example, the sense of cooperation between workers at the workplace has the greatest (and most significant) association with the probability of reporting good or very good health. However, both associations are opposite, increasing the likelihood of good health and decreasing the likelihood of very good health. All other work environment factors are also associated with a greater (lower) probability of good or very good (bad or very bad) health, except for the sense of being recognised for doing a good job, which does not show a significant association. Finally, if the employee perceives that his or her health or safety is at risk at work, or that health is affected negatively because of work, the health assessed decreases, as opposed to the increase observed when an employee perceives that health is affected positively because of work.

Results of Models 1 (GDP per capita model), 2 (Unemployment model) and 3 (Countries’ differences model) show that, after controlling for individual characteristics and work environment factors, both GDP per capita and unemployment rate associate, with opposite signs (as expected), with the probability of a certain type of self-assessed health. In addition, there are significant differences found when looking at differences in macroeconomic factors. Having a GDP above the median European GDP per capita increases the likelihood of a very bad level of self-assessed health. However, while unemployment rates above the mean associate positively with a good level of self-assessed health, they also associate negatively with very good levels of self-assessed health, compared with workers who live in countries with a worse unemployment situation.

According to our results, health is associated with monthly earnings following a U-shaped function when the level of health assessed is bad, and U-inverted when the level of self-assessed health is very good.

Finally, the countries’ differences model estimates show that, for most countries, there is a significant effect of being from a certain country over the health assessed. Although country dummies are not shown in the table, for ease of visualization of the regression results, these are included. Only for two countries (Denmark and the United Kingdom), there is a not statistically significant association.

## 5. Discussion

This article presents empirical evidence of the positive relationship between perceived health and earnings based on a model that includes a dataset from 35 European countries and 43,850 responses from active workers. As mentioned, data for the research were collected in 2015 and published in 2017. This is the most recent dataset available at the moment, allowing us to study the effect of earnings on self-assessed health while controlling with macroeconomic and microeconomic factors.

Our approach offers a quantitative analysis of the inequality of perceived health and earnings in Europe. To do so, a model to analyse the influence of earnings on different levels of self-assessed health across Europe has been conducted. This model supports the idea evidencing the positive relation between those variables based on the studied data. Furthermore, estimations of the sensitivity of earnings to age and interactions between occupation and age in the explanation of perceived health provide a more disaggregated analysis of the relationship between earnings and health compared to what is commonly offered in the literature. As an example, our results show that employees of small firms of 2–9 employees are more likely to report a very good health status compared to self-employed individuals (or firms of one worker). A behavioural explanation for this result could be the fact that workers tend to report lower health when they work more hours and are therefore overemployed, or they are working more hours than the hours that would be considered optimal [15]. Indeed, the self-employed typically report longer working hours and less time for leisure activities than wage workers [16]. Most of the small companies are family businesses which prioritise employees as a management style. Whereas some research investigated the types of situations in which the variations in health on income occur among occupations [17], this research paper complements the analysis by including age sensitivity in the different occupations. Previous research found evidence of the negative influence of the economic crisis on health [18]. Our results contradict these findings. Our model shows how higher GDP increases the likelihood of very bad self-assessed health, and respondents from countries with higher unemployment rates are more likely to report better levels of self-assessed health. This is in line with literature that has found how, in wealthier countries, patients perceive a worse impact of disease, despite having a lower objectively assessed disease activity, which means physician reported disease activity [19].

As a final note, health inequality across Europe was also tested, comparing the situation of richer and poorer countries in the European Union. As the existing literature posits, there are significant differences between countries according to their economic situation according to the effect found for unemployment. This article also confirms the large dispersion of health at work across European countries.

Regarding the methodology used, existing studies have developed a similar strategy estimating OLS regression models [11,20]. However, and despite the fact that the results are very similar, according to the nature of the dependent variable, probit models could be estimated. Although the model could be estimated as an ordered probit, we prefer to show separate bivariate probit models for each health assessed level. This is to facilitate interpretation and hypothesis of the association between each independent variable and the different levels of self-assessed health.

Understanding the dynamics of perceived health and earnings processes is important for theoretical as well as practical reasons. From a theoretical point of view, this research presents updated evidence that complements previous studies, and the results are consistent with the existing literature. From a practical perspective, these findings should contribute to health policy. because a positive and significant relationship is found between earnings and health assessed, this could be considered in the definition of economic aids and healthcare services. This is even more relevant in a context characterised by the diversity of healthcare systems in the European Union, which may also be affecting the inequality in health assessed. However, this cannot be concluded by only looking at the results from the econometric model estimated. More information would be necessary to better understand the reasons for the differences in health assessed in the different health systems. Further research on this is, therefore, highly encouraged.

As with every research study, this paper presents some limitations. Health measurement is based on data of perceived health instead of data of the real health situation. Workers surveyed answered a question related to their perceived health status (‘How is your health in general? Would you say it is...?’). Their perceptions may differ from their real health situations. The relation between the health status and other variables used in the analysis may create an endogeneity problem. Workers with better health will live for more years and they will be able to earn more money. For that reason, our results cannot be interpreted as causal effects, because of the strong correlation between earnings and health; healthier employees are able to work for longer hours, and age is also related to health. Although the aim of our model is not to demonstrate causality but to explore the association between these two variables, we understand this might be an issue and took this into consideration. Including an indicator of the GDP per capita or unemployment being above the median allows one to partially deal with the endogeneity problem, and indeed we demonstrated that this positive association between GDP and health, or the negative association between unemployment and health, is no longer true when both reach a certain threshold (the median is used in this paper, but others could be used). We encourage further research to focus on refuting this association because our results show that the direction of the association between health and earnings might not be unique.

On the other hand, these data present several advantages: the novelty of the data, collected in 2015 and published in 2017, and the possibility of including 35 countries with a high number of answers per country (more than 1500 data per country). Furthermore, as mentioned, the scales reported (from one to five) are similar to those used by other health institutions, such as the European Community Health Indicators Monitoring Joint Action. Additionally, and as was mentioned, the model only includes individual earnings from the workers who responded to the survey. The additional income of their household is not considered because the data did not provide this information. Therefore, global income should be measured because behaviours and habits are determined by the complete household income.

The research is also conditioned by the nature of the dataset. As was mentioned, no causal effects can be considered. Consequently, the development of panel data would improve this and other studies in the field. These data will allow the valuation of causal effects between the variables considered in the paper, which will offer new avenues in the field. Along this line, further research might consider real health status instead of perceived health or, moreover, the comparison between these two variables: real and self-reported health. These results could have relevant health policy implications and help to address health inequalities across Europe.

## Figures and Tables

**Figure 1 ijerph-19-00594-f001:**
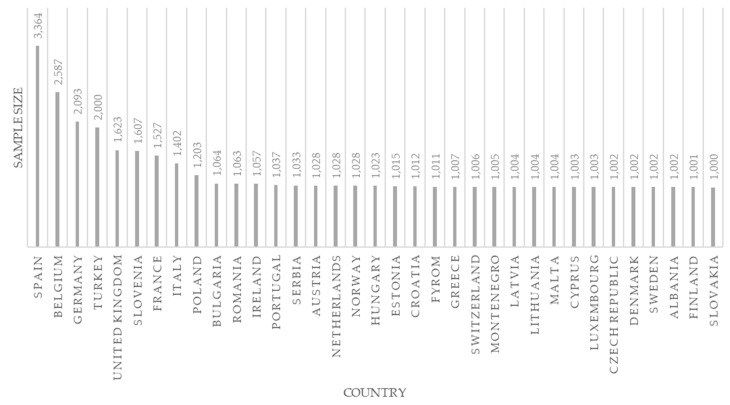
Sample sizes by country.

**Table 1 ijerph-19-00594-t001:** Summary statistics.

Variables’ Group	Variable	Obs. (*)	Mean	SD	Min	Max
Health assessed	Health assessed	43,786	4.003	0.768	1 (Very bad)	5 (Very good)
Individual factors						
Gender	Man	43,841	0.504	0.5	0	1
Age	Age	43,691	43.371	12.75	15	89
Hours educating children/grandchildren	Hours educating children/grandchildren	43,850	0.607	0.928	0	8
Marital status	Married partner	43,850	0.611	0.488	0	1
Ethnicity	Ethnicity	5755	0.349	0.477	0	1
Level of education	Early childhood education	43,689	0.006	0.076	0	1
	Primary education	43,689	0.048	0.213	0	1
	Lower secondary education	43,689	0.134	0.34	0	1
	Upper-secondary education	43,689	0.416	0.493	0	1
	Post-secondary education	43,689	0.07	0.255	0	1
	Short cycle tertiary education	43,689	0.094	0.292	0	1
	Bachelor	43,689	0.131	0.337	0	1
	Master	43,689	0.093	0.29	0	1
	Doctorate	43,689	0.009	0.097	0	1
Occupation	Elementary workers	43,850	0.104	0.305	0	1
	Plant operators	43,850	0.068	0.251	0	1
	Craft	43,850	0.118	0.322	0	1
	Skilled agricultural	43,850	0.048	0.213	0	1
	Sales workers	43,850	0.217	0.412	0	1
	Clerical	43,850	0.086	0.281	0	1
	Technicians	43,850	0.112	0.315	0	1
	Professionals	43,850	0.177	0.382	0	1
	Managers	43,850	0.063	0.243	0	1
Job characteristics						
Company size	One employee	41,653	0.126	0.331	0	1
	2–9 employees	41,653	0.245	0.43	0	1
	10–249 employees	41,653	0.361	0.48	0	1
	Over 250 employees	41,653	0.268	0.443	0	1
Increase in hours of work since job started	Increase in hours of work	43,475	2.858	0.661	1 (Increased a lot)	5 (Decreased a lot)
Work environment						
	Good job	35,053	3.917	1.059	0	1
	Conflicts are solved fairly	34,005	3.895	1.046	0	1
	Fairness	34,570	3.902	1.06	0	1
	Cooperation	34,307	4.369	0.78	0	1
	Health or safety at risk	43,050	0.251	0.434	0	1
	Health affects negatively	43,850	0.264	0.441	0	1
	Health affects positively	43,850	0.118	0.323	0	1
Macroeconomic factors						
	Unemployment rate	42,839	11.918	6.513	0	1
	GDP per capita (PPP)	42,839	30,039.37	12,979.35	9506.12	78,669.78
Monthly earnings	Monthly earnings	33,399	1346.01	2278.87	0.037	271,000

Abbreviations: Obs: Observations; SD: Standard Deviation; Min: Minimum value of variable; Max: Maximum value of variable. (*) The difference between the total number of observations in the dataset (N = 43,850) and the total number of observations in each variable is the number of missing observations for that variable.

**Table 2 ijerph-19-00594-t002:** Probit regression results for the GDP model.

Dependent Variable: Health Assessed		Very Bad	Bad	Fair	Good	Very Good
Independent Variables	Variable Name	Coef.	Coef.	Coef.	Coef.	Coef.
Intercept		−1.089	−5.933 ***	−4.857 ***	2.135 ***	−0.642
Individual factors (I)						
Gender (baseline: Women)	Man	−0.119	−0.036	−0.059 **	−0.002	0.062 **
Age (continuous)	Age	0.022 ***	0.020 ***	0.031 ***	−0.003 **	−0.033 ***
Having children (baseline: No children)	Children	−0.122 *	0.004	−0.024 **	0.036 ***	0.004
Marital status (baseline: Not married)	Married	−0.005	−0.090 **	−0.030	0.093 ***	−0.060 **
Level of education (baseline: Early childhood education)	Primary education	−0.352	−0.270	−0.064	0.207	0.099
	Lower secondary education	−0.309	−0.487 **	−0.055	0.239	0.119
	Upper secondary education	−0.655	−0.606 **	−0.073	0.194	0.230
	Post-secondary education	−0.577	−0.711 ***	0.010	0.132	0.251
	Short cycle tertiary education	−0.598	−0.716 ***	−0.164	0.142	0.388 *
	Bachelor education	−0.786	−0.765 ***	−0.095	0.124	0.341 *
	Master education	−1.082 **	−0.791 ***	−0.102	0.067	0.417 **
	Doctorate education	0.000	−0.253	−0.138	0.070	0.389 *
Type of occupation (baseline: Elementary workers)	Managers	0.673	0.399	−0.128	0.043	−0.609 ***
	Professionals	0.104	−0.010	0.117	−0.368 ***	−0.173
	Technicians	1.073 **	0.117	0.065	−0.473 ***	−0.042
	Clerical	1.046 **	0.349	0.123	−0.385 ***	−0.095
	Service and Sales workers	−0.142	−0.079	0.088	−0.388***	0.082
	Skilled agricultural	0.000	−0.331	0.196	−0.201	0.433
	Craft	1.024 **	−0.490	−0.122	−0.023	0.015
Job characteristics (JC)						
Company number of employees (baseline: 1 employee)	2–9 employees	−0.590 *	−0.253	−0.012	−0.057	0.194 *
	10–249 employees	−0.380	−0.257	0.025	−0.025	0.136
	over 250 employees	−0.212	−0.166	0.024	−0.064	0.177 *
Increase in hours worked since job started (baseline: increased a lot)	Increased a little	−0.362 *	0.052	0.042	0.036	−0.096*
	No change	−0.087	0.041	−0.063	0.049	−0.034
	Decreased a little	−0.136	0.165	−0.015	0.029	−0.059
	Decreased a lot	0.141	0.024	−0.022	−0.086	0.093
Work environment (WE)						
Work environment main factors	Good job	0.046	−0.020	−0.017	0.008	0.007
	Conflicts are solved in a fair way	−0.093 *	−0.063 **	−0.047 ***	0.003	0.055 ***
	Fairness	−0.079 *	−0.035	−0.047 ***	−0.005	0.057 ***
	Cooperation	−0.068	0.018	−0.069 ***	−0.049 ***	0.136 ***
Health or safety at risk (baseline: No)	Health or safety	0.517 ***	0.270 ***	0.123 ***	−0.104 ***	−0.051 *
Health affected because of work (baseline: No)	Health affected negatively	0.362 **	0.538 ***	0.500 ***	−0.085 ***	−0.495 ***
	Health affected positively	−0.385	0.032	−0.062 *	−0.079 **	0.125 ***
Macroeconomic factors (M)						
	Log GDP per capita (PPP)	−0.033	0.414 ***	0.332 ***	−0.233 ***	−0.035
	GDP per capita above median	0.470 **	0.105	−0.009	−0.013	0.018
Monthly earnings (E)						
	Log Monthly earnings	−0.128	0.191	0.250 **	0.058	−0.083
	(Log Monthly earnings)^2^	−0.009	−0.037 **	−0.037 ***	0.001	0.019 **
Interactions: Occupation & Age						
Type of Occupation × Age (baseline: Early Childhood education×Age)	Managers×Age	−0.004	−0.007	0.001	0.002	0.013 ***
	Professionals×Age	−0.000	0.000	−0.005 *	0.010 ***	0.005 *
	Technicians×Age	−0.017	−0.003	−0.005	0.013 ***	0.002
	Clerical×Age	−0.019 *	−0.007	−0.005	0.011 ***	0.002
	Sales×Age	0.002	0.001	−0.004	0.010 ***	−0.001
	Skilled Agricultural × Age	0.000	0.012	−0.004	0.003	−0.010
	Craft×Age	−0.026 **	0.008	0.003	0.002	−0.001
Observations		23,741	24,192	24,192	24,192	24,192
Bayesian Information Criteria (BIC)		1116.707	4606.082	20,579.19	33,546.57	25,297.30

* *p* < 0.1, ** *p* < 0.05, *** *p* < 0.001.

**Table 3 ijerph-19-00594-t003:** Probit regression results for the unemployment model.

Dependent Variable: Health Assessed		Very Bad	Bad	Fair	Good	Very Good
Independent Variables	Variable Name	Coef.	Coef.	Coef.	Coef.	Coef.
Intercept	Intercept	−0.943	−1.409 **	−1.472 **	−0.073	−1.074 **
Individual factors (I)						
Gender (baseline: Women)	Man	−0.152	−0.083 *	−0.083 ***	0.027	0.047 **
Age (continuous)	Age	0.022 ***	0.020 ***	0.030 ***	−0.003 *	−0.034 ***
Having children (baseline: No children)	Children	−0.115 *	0.002	−0.024 **	0.037 ***	0.004
Marital status (baseline: Not married)	Married	−0.022	−0.098 **	−0.039 *	0.100 ***	−0.060 **
Level of education (baseline: Early childhood education)	Primary education	−0.395	−0.320	−0.160	0.217	0.162
	Lower secondary education	−0.379	−0.573 **	−0.199	0.256 *	0.214
	Upper secondary education	−0.719	−0.724 ***	−0.246	0.240 *	0.321
	Post-secondary education	−0.665	−0.812 ***	−0.116	0.175	0.285
	Short cycle tertiary education	−0.658	−0.846 ***	−0.328 *	0.203	0.441 **
	Bachelor education	−0.884 *	−0.936 ***	−0.282 *	0.214	0.392 *
	Master education	−1.184 **	−0.960 ***	−0.295 *	0.149	0.477 **
	Doctorate education	0.000	−0.441	−0.321	0.160	0.436 **
Type of occupation (baseline: Elementary workers)	Managers	0.576	0.321	−0.190	0.079	−0.585 **
	Professionals	0.052	−0.023	0.106	−0.364 ***	−0.170
	Technicians	1.023 **	0.098	0.049	−0.469 ***	−0.031
	Clerical	1.061 **	0.371	0.127	−0.382 **	−0.096
	Service and Sales workers	−0.099	−0.016	0.115	−0.401 ***	0.078
	Skilled agricultural	0.000	−0.319	0.195	−0.190	0.410
	Craft	0.950 **	−0.551 *	−0.161	−0.006	0.019
Job characteristics (JC)						
Company number of employees (baseline: 1 employee)	2–9 employees	−0.557 *	−0.245	−0.019	−0.055	0.201 *
	10–249 employees	−0.390	−0.267 *	0.001	−0.019	0.157
	over 250 employees	−0.218	−0.167	0.004	−0.072	0.210 **
Increase in hours worked since job started (baseline: increased a lot)	Increased a little	−0.369 *	0.051	0.044	0.032	−0.096 *
	No change	−0.096	0.039	−0.055	0.049	−0.043
	Decreased a little	−0.115	0.203	0.015	0.003	−0.061
	Decreased a lot	0.237	0.130	0.044	−0.138 *	0.090
Work environment (WE)						
Work environment main factors	Good job	0.047	−0.023	−0.022*	0.009	0.010
	Conflicts are solved in a fair way	−0.100 **	−0.069 **	−0.049 ***	0.004	0.055 ***
	Fairness	−0.071	−0.029	−0.047 ***	−0.007	0.059 ***
	Cooperation	−0.069	0.022	−0.062 ***	−0.051 ***	0.132 ***
Health or safety at risk (baseline: No)	Health or safety	0.512 ***	0.261 ***	0.127 ***	−0.097 ***	−0.066 **
Health affected because of work (baseline: No)	Health affected negatively	0.334 **	0.535 ***	0.507 ***	−0.084 ***	−0.510 ***
	Health affected positively	−0.352	0.066	−0.055	−0.099 ***	0.144 ***
Macroeconomic factors (M)						
	Unemployment rate	−0.018 *	−0.016 **	−0.020 ***	0.005 **	0.015 ***
	Unemployment above median	0.044	0.087	0.063 **	−0.115 ***	0.071 **
Monthly earnings (E)						
	Log Monthly earnings	−0.269 **	0.025	0.274 **	0.090	−0.187 *
	(Log Monthly earnings)^2^	0.013	−0.011	−0.031 ***	−0.010	0.030 ***
Country factors						
	Country dummies	No	No	No	No	No
Interactions: Occupation & Age						
Type of Occupation×Age (baseline: Early childhood education×Age)	Managers×Age	−0.004	−0.007	0.001	0.001	0.013 **
	Professionals×Age	−0.000	−0.000	−0.005 *	0.011 ***	0.005 *
	Technicians×Age	−0.016	−0.003	−0.005	0.013 ***	0.002
	Clerical×Age	−0.019 *	−0.008	−0.005	0.011 ***	0.002
	Sales × Age	0.002	−0.000	−0.004 *	0.010 ***	−0.001
	Skilled Agricultural × Age	0.000	0.013	−0.003	0.003	−0.010
	Craft × Age	−0.025 **	0.009	0.003	0.001	−0.001
Observations		23,741.000	24,192.000	24,192.000	24,192.000	24,192.000
Bayesian Information Criteria (BIC)		1124.525	4640.259	20,573.364	33,594.765	25,154.170

* *p* < 0.1, ** *p* < 0.05, *** *p* < 0.001.

**Table 4 ijerph-19-00594-t004:** Probit regression results for the countries’ differences model.

Dependent Variable: Health Assessed		Very Bad	Bad	Fair	Good	Very Good
Independent Variables	Variable Name	Coef.	Coef.	Coef.	Coef.	Coef.
Intercept	Intercept					
Individual factors (I)						
Gender (baseline: Women)	Man	−0.143	−0.022	−0.042 *	−0.020	0.072 ***
Age (continuous)	Age	0.027 ***	0.022 ***	0.029 ***	−0.002	−0.034 ***
Having children (baseline: No children)	Children	−0.133 **	0.007	−0.016	0.035 ***	−0.003
Marital status (baseline: Not married)	Married	−0.022	−0.087 **	−0.026	0.085 ***	−0.045 **
Level of education (baseline: Early childhood education)	Primary education	−0.385	−0.243	0.009	0.141	0.180
	Lower secondary education	−0.338	−0.441 **	−0.101	0.286 *	0.112
	Upper secondary education	−0.691	−0.617 **	−0.123	0.266 *	0.194
	Post-secondary education	−0.319	−0.695 **	−0.161	0.208	0.287
	Short cycle tertiary education	−0.516	−0.653 **	−0.232	0.222	0.337 *
	Bachelor education	−0.752	−0.775 ***	−0.165	0.195	0.313
	Master education	−1.027	−0.779 ***	−0.184	0.146	0.387 *
	Doctorate education	0.000	−0.238	−0.224	0.129	0.385 *
Type of occupation (baseline: Elementary workers)	Managers	0.736	0.516	−0.251	−0.016	−0.456 **
	Professionals	0.040	0.028	0.123	−0.336 ***	−0.231 **
	Technicians	1.198 **	0.116	−0.019	−0.444 ***	−0.034
	Clerical	1.210 **	0.424	0.116	−0.356 **	−0.143
	Service and Sales workers	0.011	−0.052	0.081	−0.339 ***	0.021
	Skilled agricultural	0.000	−0.390	0.196	−0.266	0.448
	Craft	1.232 **	−0.475	−0.186	−0.026	0.058
Job characteristics (JC)						
Company number of employees (baseline: 1 employee)	2–9 employees	−0.629 *	−0.212	−0.032	−0.016	0.144
	10–249 employees	−0.413	−0.226	−0.033	0.012	0.119
	over 250 employees	−0.248	−0.113	−0.011	−0.039	0.152
Increase in hours worked since job started (baseline: Increased a lot)	Increased a little	−0.415 *	0.038	0.012	0.028	−0.057
	No change	−0.130	0.053	−0.104 **	0.032	0.020
	Decreased a little	−0.123	0.178	−0.034	0.034	−0.050
	Decreased a lot	0.082	0.033	−0.006	−0.069	0.063
Work environment (WE)						
Work environment main factors	Good job	0.065	−0.005	−0.023 **	0.008	0.008
	Conflicts are solved in a fair way	−0.105 **	−0.079 ***	−0.043 ***	0.001	0.057***
	Fairness	−0.077 *	−0.037	−0.047 ***	−0.007	0.064 ***
	Cooperation	−0.087	0.018	−0.075 ***	−0.049 ***	0.140 ***
Health or safety at risk (baseline: No)	Health or safety	0.523 ***	0.271 ***	0.134 ***	−0.114 ***	−0.044
Health affected because of work (baseline: No)	Health affected negatively	0.365 **	0.562 ***	0.491 ***	−0.088 ***	−0.494 ***
	Health affected positively	−0.376	0.043	−0.068 *	−0.081 **	0.139 ***
Monthly earnings (E)						
	Log Monthly earnings	−0.235	0.188	0.161	0.041	−0.037
	(Log Monthly earnings)^2^	−0.002	−0.037 **	−0.026 **	0.003	0.013 *
Country factors						
	Country dummies	YES	YES	YES	YES	YES
Interactions: Occupation & Age						
Type of Occupation×Age (baseline: Early childhood education×Age)	Managers×Age	−0.007	−0.009	0.002	0.002	0.011 **
	Professionals×Age	−0.002	−0.001	−0.006 **	0.010 ***	0.006 **
	Technicians×Age	−0.021 *	−0.003	−0.003	0.012 ***	0.003
	Clerical×Age	−0.023 **	−0.009	−0.004	0.010 ***	0.003
	Sales×Age	−0.001	0.000	−0.003	0.009 ***	0.000
	Skilled Agricultural × Age	0.000	0.014	−0.004	0.004	−0.010
	Craft × Age	−0.030 ***	0.008	0.004	0.002	−0.002
Observations		17,888	24,736	24,736	24,736	24,736
Bayesian Information Criteria (BIC)		1253.438	4939.483	20,830.773	34,231.084	25,238.926

* *p* < 0.1, ** *p* < 0.05, *** *p* < 0.001.

## Data Availability

The data that support the findings of this study are available in the EWCS (European working conditions survey) from the European Foundation for the Improvement of Living and Working Conditions (https://www.eurofound.europa.eu/). Data are available upon request.

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
