# Peer review of "Perceived Health and Earnings: Evidence from the European Working Conditions Survey 2015"

_ijerph, 2022, doi:10.3390/ijerph19010594_

Round 1

Reviewer 1 Report

This is an interesting, well written article on exploring the relationship between perceived health and earnings in Europe. 

The statistical methods used are relatively straight forward (linear regression and t-tests). Given the large sample size, it is not surprising that many quantities are significant. In a study like this, graphical representations of the data may tell some interesting stories and the paper does not have any graphics at all.  One thing to consider would be to add some graphical comparisons instead of the t-tests to show the differences (or lack thereof).

The authors mention some limitations, but do not provide any guidance on how to address them. Some sensitivity analyses to show robustness to unmeasured confounding and other forms of bias would also strengthen the paper.

Author Response

Reviewer 1 comments

This is an interesting, well written article on exploring the relationship between perceived health and earnings in Europe. 

General response: We thank to the reviewer for the useful comments that, we believe, have significantly improved the quality of the paper. Please, note that, given your comments and the comments from a second reviewer, we had to include more tables. However, this is to facilitate the comprehension of the results. In addition, it helps to demonstrate the robustness of our results.

We now respond separately to each of your comments.

Please note that, when we refer to new text / text modified (compared to the previous version) we will refer to lines in ‘simple revisions’ mode (that means, hiding the track changes).

Comment #1: The statistical methods used are relatively straight forward (linear regression and t-tests). Given the large sample size, it is not surprising that many quantities are significant. In a study like this, graphical representations of the data may tell some interesting stories and the paper does not have any graphics at all.  One thing to consider would be to add some graphical comparisons instead of the t-tests to show the differences (or lack thereof).

Response: Thank you for your comment. We have included a figure in the results section, at the beginning. The figure explains the sample size of the EWCS 2015 and we believe it is informative. We find that the rest of results, especially the regression models, are better explained with tables as they allow for hypothesis testing.

Comment #2: The authors mention some limitations, but do not provide any guidance on how to address them. Some sensitivity analyses to show robustness to unmeasured confounding and other forms of bias would also strengthen the paper.

Response: Thank you for this comment. We have assumed robust standard errors in our estimates (we have included this in the text in line 137: “Robust standard errors are assumed in all our estimations”). In addition, we have estimated bivariate probit models, so our dependent variable is now disaggregated in its 5 original categories. On the one hand, this provides two pieces of information: on the one hand, results are very similar to those obtained through OLS. This shows our model is robust. On the other hand, results are much more informative now, and allow for more hypothesis to be tested (the association between each independent variable with each level of health assessed).

Reviewer 2 Report

This paper uses the European Working Conditions Survey to investigate the association between earnings and health.

This paper needs to be checked thoroughly, not only for English but also for content and layout. Table 2 is unreadable – it is not clear, beyond the first couple of columns of what belongs to what. Lines 240 – 243 are either instructions to authors or are previous reviewers’ comments. This should not be here. The title also doesn’t make sense.

As for the research itself, I am confused about what the paper is exactly trying to do. Is this just updating the association between earnings and health? What is the novelty here? The data are from 2015 and so are already reasonably old by this point. This may be the most recent survey available but it is not really that up-to-date.

Furthermore, the data only seem to come from individuals who are working, not those people out of the labour force. How do you consider those who aren’t working? There is no sample selection model here that I can see. You would need to at least consider this issue and have more of a discussion of the endogeneity issues. You note in the discussion that endogeneity may be a problem but I think this needs to be more in-depth.

Does your health outcome variable mean the same thing across different countries? This has been a problem with Europe wide data in the past. There are different ways of answering the same question. If you look at work by Jones, Rice and Robone on anchoring vignettes using SHARE data you will see this.

I am have number of concerns about the modelling. Your outcome variable is ordered and yet you estimate linear regression models. I am not averse to the use of linear regression models with limited dependent variables. However, I think it would be better to dichotomise the outcome variable and then estimate linear probability models. These at least have a clear interpretation. Currently your results are very hard to interpret.

A number of your independent variables are also ordered but you include them as continuous variables. You should turn these variables into a series of dummy variables and include them that way. For example, the job hours variable is very hard to interpret. What does the mean represent here?

Earnings should be logged – this again would help with interpretation – as should GDP.

You have an unemployment rate average of 12% this seems very high for what are mostly EU countries. Are these sample statistics weighted?

Why not simply use country fixed effects and have the results as withing country differences?

I am not sure what Table 3 represents.

You cite Wang et al 2018 in the text but it isn’t in the references.

Author Response

Reviewer 2

This paper uses the European Working Conditions Survey to investigate the association between earnings and health.

General response: We thank to the reviewer for the useful comments that, we believe, have significantly improved the quality of the paper. In particular, the suggestion of estimating probit models, instead of OLS, has brought too many interesting results, otherwise hidden. Results do not particularly vary, but now not only interpretation is easier, but also the models are much more informative. We had to include more tables, but it does, indeed, facilitate the comprehension of the results. In addition, it helps to demonstrate the robustness of our results.

We now respond separately to each of your comments.

Please note that, when we refer to new text / text modified (compared to the previous version) we will refer to lines in ‘simple revisions’ mode (that means, hiding the track changes).

Comment #1. This paper needs to be checked thoroughly, not only for English but also for content and layout. Table 2 is unreadable – it is not clear, beyond the first couple of columns of what belongs to what. Lines 240 – 243 are either instructions to authors or are previous reviewers’ comments. This should not be here. The title also doesn’t make sense.

Response to Comment #1:

Thank you for this comment.

We have reviewed the English, content and layout. Because the deadline and dates we have not been able to find a person to formally review and do the English editing. However, we have someone that could do the English editing. If you still think there is need of reviewing the English after reading the resubmitted version of the paper, we will be happy to send the paper to this person.

Table 2 has been corrected in a way so that, we believe, it is self-explanatory now. We have included an additional column with the variable groups and the name of the variables in another column.

Regarding lines 240-243, we are sorry about that. Indeed, this should not be there. It was part of the text of the paper template, and it has been removed.

We have changed the Manuscript title. The new title is: “Perceived health and earnings: Evidence from the European Working Conditions Survey 2015.”

Comment #2: As for the research itself, I am confused about what the paper is exactly trying to do. Is this just updating the association between earnings and health? What is the novelty here? The data are from 2015 and so are already reasonably old by this point. This may be the most recent survey available but it is not really that up-to-date.

Response: Thank you for this comment. The EWCS 2015 is the most recent available data including health information for the population of active workers. We have added a comment regarding this in the text, at lines Although the 2021 survey has been done, this information will not be available, we believe, until 2023.

The main novelty of this paper is that we incorporate macroeconomic and microeconomic indicators in the same model. In other words, we control by macroeconomic indicators when trying to explain the association between health and earnings within the active population of workers. In addition, because the EWCS includes only people actively employed, we also explore differences in the health assessed based on microeconomic indicators, such as the type of occupation, as well as based on GDP and unemployment groups being above versus below the median of European countries.

Comment #3: Furthermore, the data only seem to come from individuals who are working, not those people out of the labour force. How do you consider those who aren’t working? There is no sample selection model here that I can see. You would need to at least consider this issue and have more of a discussion of the endogeneity issues. You note in the discussion that endogeneity may be a problem but I think this needs to be more in-depth.

Response: Thank you for this comment. The EWCS only interviews individuals who are working at the time of the interview. Therefore, there is no way we can compare the association of health and earnings between those who are working with those who are not.

At lines 50-52 we have reviewed the text: “This questionnaire collected demographic data and a broad range of factors about current working conditions. The specificity of this dataset is that it includes the population of active workers only.”

However, we include occupation type, so that the model allows testing differences in the health assessed by occupation type, instead of just providing a global health assessment for the population of workers. In addition, we conducted t-tests to complement the regression results so that we can compare health assessed and other variables between populations for macroeconomic indicators being below or above the mean.

In section 2.2.5 Macroeconomic factors (M), at lines 120-125, we have added new text to clarify this:

“We also create a dichotomous variable to distinguish when GDP/unemployment of a country are above the median versus when its GDP/unemployment is below the median. Because the country GDP per capita, the unemployment rate of a country and the country are linearly dependent, including more than one of these variables would result in a mis-specified model, because of perfect multicollinearity. Separate models are, thus, estimated.”

Finally, we have included some additional thoughts in the discussion regarding the endogeneity problems between health and earnings.

At lines 306-314, we added the following text: “Although the aim of our model is not to demonstrate causality but to explore the association between these two variables, we understand this might be an issue, and took this into consideration. Including an indicator of the GDP per capita or unemployment being above the median allow to partially deal with the endogeneity problem, and indeed we demonstrated that this positive association between GDP and health, or the negative association between unemployment and health, is no longer true when both reach a certain threshold (median is used in this paper but other could be used). We encourage further research to focus on refuting this association, as our results show the direction of the association between health and earnings might not be unique.”

Comment #4: Does your health outcome variable mean the same thing across different countries? This has been a problem with Europe wide data in the past. There are different ways of answering the same question. If you look at work by Jones, Rice and Robone on anchoring vignettes using SHARE data you will see this.

Response: Thank you for this comment. We are not sure to understand your concern/issue. The EWCS is asking the same question in every country, and therefore it is assumed that, given the Eurofound is providing a unique dataset with a unique variable for health status (for all the countries), these should be comparable and have the same meaning. Independently of the language of the questionnaire, it is true that the threshold for a very good health might be different, but so might it be for individuals who speak the same language and live in the same country. The subjectiveness of the instrument is, therefore, a limitation. But this is already mentioned in the discussion section.

Comment #5: I am have number of concerns about the modelling. Your outcome variable is ordered and yet you estimate linear regression models. I am not averse to the use of linear regression models with limited dependent variables. However, I think it would be better to dichotomise the outcome variable and then estimate linear probability models. These at least have a clear interpretation. Currently your results are very hard to interpret.

Response: Thank you for this comment. We have replaced OLS estimation by probit models. This results in a much more informative regression analysis, although the number of tables in the manuscript also is higher. We present three regression tables (Tables 2-4): the GDP model, the unemployment model, and the countries’ model.

Comment #6: A number of your independent variables are also ordered but you include them as continuous variables. You should turn these variables into a series of dummy variables and include them that way. For example, the job hours variable is very hard to interpret. What does the mean represent here?

Response: Thank you. We have included the working hours as a categorical variable. This variable represents the worker perception with the increase in hours worked since the job started.

Earnings should be logged – this again would help with interpretation – as should GDP.

Response: Thank you, we have generated the log of earnings and GDP.

You have an unemployment rate average of 12% this seems very high for what are mostly EU countries. Are these sample statistics weighted?

Why not simply use country fixed effects and have the results as withing country differences?

Response: Thank you for this comment. The countries’ model indeed includes country dummies, as fixed effects. It still does now that we have changed from OLS to probit. We though about your comment and replaced the heterogeneity analysis in the previous version of the paper (previous Table 3) by including in the GDP and unemployment models a dichotomous variable indicating whether the macroeconomic indicator is above or below the median value for the European countries. 

I am not sure what Table 3 represents.

Response: Thank you. As explained in the previous response this has now been removed.

You cite Wang et al 2018 in the text but it isn’t in the references.

Thank you. We have incorporated the missing reference.
